# The Contribution of Postprandial Glucose Levels to Hyperglycemia in Type 2 Diabetes Calculated from Continuous Glucose Monitoring Data: Real World Evidence from the DIALECT-2 Cohort

**DOI:** 10.3390/nu16203557

**Published:** 2024-10-20

**Authors:** Niala den Braber, Miriam M. R. Vollenbroek-Hutten, Sacha E. M. Teunissen, Milou M. Oosterwijk, Kilian D. R. Kappert, Gozewijn D. Laverman

**Affiliations:** 1Division of Nephrology, Department of Internal Medicine, Ziekenhuisgroep Twente, 7609 PP Almelo, The Netherlands; s.e.m.teunissen@utwente.nl (S.E.M.T.); g.laverman@zgt.nl (G.D.L.); 2Biomedical Signals and Systems, Faculty of Electrical Engineering, Mathematics and Computer Science, Technical Medical Centre, University of Twente, 7522 NB Enschede, The Netherlands; m.m.r.hutten@utwente.nl; 3Biomedical Photonic Imaging, Faculty of Science and Technology, Technical Medical Centre, University of Twente, 7522 NB Enschede, The Netherlands; 4Department of Surgery, Ziekenhuisgroep Twente, 7609 PP Almelo, The Netherlands

**Keywords:** Type 2 diabetes, continuous glucose monitoring (CGM), postprandial plasma glucose, fasting plasma glucose, hyperglycemia, glucose management, Hemoglobin A1c (HbA_1c_), carbohydrates

## Abstract

Background/Objectives: Traditional glycemic monitoring in type 2 diabetes is limited, whereas continuous glucose monitoring (CGM) offers better insights into glucose fluctuations. This study aimed to determine the correlations and relative contributions of fasting plasma glucose (FPG) and postprandial plasma glucose (PPG) levels to hyperglycemia. Methods: We utilized CGM and recorded carbohydrate intake data from lifestyle diaries of 59 patients enrolled in the Diabetes and Lifestyle Cohort Twente (DIALECT-2). Correlations between FPG and the glucose management indicator (GMI), FPG and Time Above Range (TAR), PPG and GMI, and PPG and TAR were conducted. Daily and mealtime relative contributions of PPG and FPG to glycated hemoglobin (HbA_1c_) and GMI were determined, considering two ranges: on target (<7.0%, 53 mmol/mol) and not on target (≥7.0%, 53 mmol/mol). Correlations between mealtime PPG and carbohydrate consumption were examined. Results: FPG and PPG correlated with GMI (r = 0.82 and 0.41, respectively, *p* < 0.05). The relative contribution of PPG in patients with HbA_1c_, GMI, and TAR values not on target was lower than in patients with HbA_1c_, GMI, and TAR values on target. When analyzing different mealtimes, patients with target GMI values had a higher PPG (73 ± 21%) than FPG after breakfast (27 ± 21%, *p* < 0.001). Individuals with elevated GMI levels had lower PPG after lunch (30 ± 20%), dinner (36 ± 23%), and snacks (34 ± 23%) than FPG. PPG after breakfast positively correlated (r = 0.41, *p* < 0.01) with breakfast carbohydrate intake. Conclusions: Both PPG and FPG contribute to hyperglycemia, with PPG playing a larger role in patients with better glycemic control, especially after breakfast. Targeting PPG may be crucial for optimizing glucose management.

## 1. Introduction

Reducing hyperglycemia is of utmost importance in mitigating the morbidity and mortality associated with type 2 diabetes mellitus. The traditional approach to monitoring glycemic control, i.e., the assessment of glycated hemoglobin (HbA_1c_) measurements in combination with self-measurements of blood glucose (SMBG) by fingerpricking, is imperfect at accurately capturing hyperglycemic episodes. HbA_1c_ only indicates average blood glucose levels, whereas SMBG yields unsystematic and non-analyzable data [1,2,3].

Continuous glucose monitoring (CGM) technology provides the opportunity for a deeper understanding of glycemic control despite its limited availability in clinical practice for patients with type 2 diabetes. CGM-derived metrics, such as time in range (TIR), time above range (TAR), time below range (TBR), and the glucose management indicator (GMI), provide valuable insights [4,5,6]. The GMI reflects the average blood glucose levels, acting as a CGM-based HbA_1c_ equivalent [7].

Importantly, CGM also provides the opportunity to evaluate the impact of meals on hyperglycemia by distinguishing the fasting plasma glucose (FPG) values and the plasma glucose levels after meals, i.e., the postprandial plasma glucose (PPG). PPG concentrations provide insight into the impact of carbohydrate intake on glucose levels, and this enables a more detailed evaluation of glucose responses after different meals or specific mealtimes. By determining the relative contribution of PPG and FPG to TAR, this approach aids in setting treatment priorities, whether to address PPG or FPG first. Identifying the primary factor contributing to hyperglycemia allows for clear treatment goals, empowering patients to manage their glucose levels through dietary changes or insulin dose adjustments [8]. Consequently, distinguishing the relative contribution of PPG and FPG can help reduce TAR and thus improve TIR [9,10,11,12]. 

This observational real-world study aims to determine the correlations and relative contributions of FPG and PPG to overall hyperglycemia in patients with type 2 diabetes. Additionally, we aim to assess these contributions after different meals to gain a comprehensive understanding by analyzing the differences in the relative contribution of FPG and PPG in patients with type 2 diabetes.

## 2. Materials and Methods

### 2.1. Patient Inclusion and Data Collection

In this study, we used data from all patients enrolled in the DIAbetes and LifEstyle Cohort Twente (DIALECT)-2 between March 2017 and May 2019. DIALECT is an observational study with the inclusion criteria of being an adult with type 2 diabetes, receiving treatment at Ziekenhuisgroep Twente (ZGT) hospital, Almelo, and Hengelo, the Netherlands. The exclusion criteria included end-stage kidney disease and an inability to comprehend the informed consent procedure. Detailed descriptions of the study procedures for DIALECT have been published earlier [13,14]. The study is registered at the International Clinical Trials Registry Platform (ICTRP) under the trial code NTR5855, accessible at https://trialsearch.who.int/ (accessed on 15 October 2024).

### 2.2. Ethical Considerations

The study was performed in accordance with the Declaration of Helsinki and the guidelines of good clinical practice. The Medical Research Ethics Committees United (MEC-U) in Nieuwegein, The Netherlands, with the registration number NL57219.044.16, reviewed and approved the protocol. Before participating, all patients provided informed consent by signing the appropriate documents.

For this study, de-identified data of the DIALECT-2 study were used, ensuring the confidentiality and privacy of participant information. Participants in this study were provided financial compensation for travel and parking expenses if necessary.

### 2.3. Data Collection

Baseline registration for DIALECT-2 involved three hospital visits with a one-week interval between each visit. Patients were instructed to maintain their usual lifestyle habits during this period. At the baseline visit, we obtained information on age, diabetes duration, microvascular and macrovascular diseases, and medication use from electronic patient files, which was verified with the patients.

To collect intermittently scanned CGM (isCGM) data, we used a FreeStyle Libre^®^ sensor (Abbott Diabetes Care, Alameda, CA, USA), which was applied to the patient’s upper arm. The sensor measured individual glucose levels for a duration of 2 weeks and stored up to 8 h of data. Data from the sensor were stored using the FreeStyle Libre reader, which was blinded for the patients using a custom-made 3D printed case. Subsequently, the data from the reader were uploaded in MATLAB^®^ (2022a, The MathWorks, Inc., Natick, MA, USA). To be considered valid, patients’ data required at least 3 days of measurements, each with at least 90% of data availability. The TAR was calculated based on the target glycemic range defined as above 10.0 mmol/L, and the TAR level 2 (TAR2) represented the target glycemic range above 13.9 mmol/L [4,6].

To assess dietary intake, patients kept a detailed food diary over a two-week period. They recorded the time of consumption, types and sources of food items, and the weight or volume of each item. The number of servings was expressed in natural units (e.g., slices of bread), household measures (e.g., cups or spoons), or weight-based measurements. For natural units and household measures, an average serving size was used. Dietary data were then translated into daily nutrient intake utilizing the Dutch Food Composition Database, 2016 edition [15]. Compliance of the food diary entries was determined by the time and meal information provided. Compliance of the principal meals (breakfast, lunch, and dinner) was categorized as either compliant (100%) or non-compliant (0%). The mean of these three values constituted the overall daily adherence rate. For the methodology used to evaluate compliance, see Appendix A. Patients who maintained at least 80% compliance on average over the two-week period were included.

### 2.4. Data Analysis

Data from the FreeStyle Libre Reader were saved to a text file and preprocessed with MATLAB. The sensor recorded data at 15 min intervals on average, varying from 13 to 19 min. Readings were resampled to one sample every 15 min using linear interpolation, marking any interpolated data point as missing if the interval between the original data points exceeded 30 min.

For each patient, daily fasting plasma glucose (FPG) levels were determined as the glucose concentration 15 min prior to breakfast, deemed valid if preceded by a minimum 8 h fasting period and if breakfast fell between 5:00 and 10:00 a.m. Both daily postprandial plasma glucose (PPG) and FPG were quantified from 7:00 to 22:00, with post-meal contribution calculated 120 min after each meal, a time window recommended by the American Diabetes Association [16].

PPG and FPG contributions were assessed by calculating the incremental area under the curve (iAUC) above baseline FPG levels (iAUC1) for daily PPG, and above 6.1 mmol/L (110 mg/dL, iAUC2), the upper limit of normal preprandial plasma glucose defined by the American Diabetes Association, to reflect combined FPG and PPG increases [16,17,18]. The FPG contribution was derived as the difference between iAUC2 and iAUC1 [10]. These calculations are depicted in Figure 1.

The respective contributions of PPG and FPG to overall glucose levels were calculated using (iAUC1/iAUC2) × 100 for PPG, and (iAUC2-iAUC1)/iAUC2 × 100 for FPG. Mean iAUC1 and iAUC2 values, alongside relative PPG and FPG, were computed for each patient.

The food intake dataset included dates, times, and carbohydrate amounts ingested in grams per meal. To assess the relevance of carbohydrates per meal, we analyzed the glucose data between the time food was consumed and 2 h afterward. Several parameters were calculated, including the slope (mmol/L/min), peak (mmol/L), peak time (min), ∆glucose, and intersection with the *y*-axis.

To determine which meals should be considered in the analysis, we needed to establish the minimum carbohydrate intake that triggers a rise in glucose levels. We assumed that a rise of 0.5 mmol/L after consumption was attributable to the carbohydrate intake of that meal, snack, or beverage. Utilizing the baseline intersection and slope, the minimum significant carbohydrate quantity per patient was determined. Intakes starting from a minimal threshold of 10.1 g of carbohydrates began to affect plasma glucose concentrations; thus, meal instances with carbohydrate consumption below 10 g were excluded from the analysis.

Meal timing effects on PPG and FPG were studied by eliminating overlapping postprandial periods through specific assumptions. A 2 h postprandial window was adopted [11]. Intakes within 60 min of a preceding meal were merged; those between 60 and 105 min led to exclusion from analysis; intakes occurring between 105 and 120 min after a prior meal resulted in inclusion. Meal categorizations were set as follows: 06:00–09:00 for breakfast, 12:00–14:00 for lunch, 17:00–19:00 for dinner, and other times as snacks, in line with typical Dutch mealtimes [19]. These criteria facilitated an accurate assessment of PPG and FPG across different mealtimes while accounting for potential overlap and meal intervals.

### 2.5. Statistical Analysis

Variables with normal distributions, as assessed by frequency histogram visual inspection, were presented as mean ± standard deviation (SD). Variables with skewed distributions were described using the median and interquartile range (IQR), while dichotomous and categorical variables were summarized by counts and percentages.

Pearson’s correlation coefficient (r) was calculated to evaluate the relationship between HbA_1c_, GMI, and TAR with both PPG and FPG, aiming to determine if PPG and FPG parameters reflect GMI, TAR, and HbA_1c_. Additionally, the correlation between carbohydrate intake and PPG was assessed.

The relative contributions of fasting and daily PPG to the total glucose increments were compared across two groups based on GMI, TAR, and HbA_1c_ targets. The analysis was conducted across two groups to determine if patients with target glycemic values have different treatment focuses than those with elevated glycemia. For GMI and HbA_1c_, groups were categorized by achievement of target values of <7.0% (53 mmol/mol) or not [16]. For TAR, groups were classified by attainment of a threshold (<25%) or not [4]. Group differences were analyzed using one-way ANOVA with Tukey’s post hoc test for normally distributed variables and the Kruskal–Wallis’s test followed by Dunn’s post hoc test for variables with skewed distributions.

Furthermore, differences between mealtime iAUC1, ∆glucose, slope, peak time, peak glucose levels, and carbohydrate intake post-breakfast, lunch, dinner, and snacks were compared using either one-way ANOVA or Kruskal–Wallis’s test, complemented by post hoc Dunn’s test.

A *p*-value of less than 0.05 was considered statistically significant across all tests to account for the risk of Type I error in secondary analyses, Holm-Bonferroni corrections were employed, with the significance level set at α = 0.05. All statistical procedures were conducted using MATLAB.

### 2.6. Data and Resource Availability

The dataset analyzed in this study is not yet publicly available due to ongoing participant inclusion but can be obtained from the principal investigator upon reasonable request.

## 3. Results

A total of 87 patients were included in the DIALECT-2 cohort who used the FreeStyle Libre sensor and registered their food intake for 2 weeks. Of these patients, 61 patients had a compliance of >80% for the food diaries. Two people did not have sufficient glucose data and were excluded. As a result, a total of 59 patients had sufficient glucose and food intake data. The mean age of these patients was 65.7 ± 10.0 years and two-thirds of the patients were men (66%). Mean diabetes duration was 14.2 ± 10.7 years, mean HbA_1c_ 7.4 ± 3.0% (56.9 ± 9.1 mmol/mol), mean BMI 31.0 ± 5.2 kg/m^2^, and 59% (35 persons) used insulin, of which 71% (25 people) were on a multiple daily injection regimen (Table 1).

### 3.1. FPG and Daily PPG

The mean FPG for all patients was 7.0 ± 1.8 mmol/L, with 20.3% of patients maintaining a median FPG below 5.6 mmol/L. All correlations between PPG and FPG with TAR, TAR2, GMI, and HbA_1c_ were significant. Strong correlations were observed between FPG and TAR (r = 0.80, *p* < 0.001) and between FPG and GMI (r = 0.82, *p* < 0.001), indicating that higher FPG values are associated with increased TAR and GMI. The correlations between PPG and TAR, TAR2, GMI, and HbA_1c_ were modest (r = 0.33–0.41, *p* < 0.05), which were lower than those observed between FPG and TAR, TAR2, GMI and HbA_1c_ (r = 0.48–0.82, *p* < 0.001).

While the relative contribution of daily PPG and FPG to overall glucose levels was roughly 2/3 versus 1/3 in the total population (65.4 ± 26.6% versus 34.6 ± 26.6%), this was different depending on the level of glycemic control, as depicted in Figure 2A. In the group of patients with HbA_1c_ < 7.0% (53 mmol/mol), the relative contribution of PPG was higher than in patients with HbA_1c_ ≥ 7.0% (53 mmol/mol) (76.3 ± 23.4% versus 60.6 ± 26.8%, *p* < *0*.05). Similarly, the group of patients with GMI < 7.0% (53 mmol/mol) had a higher relative contribution of PPG than patients with GMI ≥ 7.0% (53 mmol/mol) (75.9 ± 23.9% versus 49.2 ± 22.6%, *p* < 0.001), as depicted in Figure 2B.

A similar trend was observed with TAR levels; the relative contribution of PPG was less in individuals with high TAR values (52.7 ± 23.7%) than in those with TAR values below the 25% consensus target (73.4 ± 25.5%, *p* = 0.003).

### 3.2. Mealtime Postprandial Plasma Glucose

A median of 2.1% of the consumptions containing at least 10 g of carbohydrates occurred within 60 min of a preceding meal and were therefore combined. A median 6.8% of the meals occurred within 60–105 min after a prior meal and were consequently omitted. After the combination or removal of these overlapping eating events, there remained a total of 2774 meal and snack instances, averaging 3.2 events per patient per day.

Comparison across the types of meals, including breakfast, lunch, dinner, or snacks, revealed no statistically significant differences in the total postprandial glucose. However, breakfast resulted in the highest average increase in glucose levels (mean Δglucose, 3.6 ± 1.4 mmol/L; slope, 0.05 ± 0.02), surpassing lunch, dinner, and snack times, which demonstrated mean Δ glucose values of 2.5 ± 1.6, 2.4 ± 1.2, and 2.0 ± 1.1 mmol/L, with slopes of 0.03 ± 0.02, 0.03 ± 0.02, and 0.02 ± 0.02, respectively (Table 2). 

Patients with a GMI above the target range exhibited a lower PPG and higher FPG across all meal types when compared to those with target GMI levels (as illustrated in Figure 3). Notably, patients within the target GMI range experienced a greater PPG contribution during breakfast (73 ± 21%) relative to the contribution of FPG (27 ± 21%, *p* < 0.001). Conversely, no significant variance was observed between PPG and FPG contributions in other meals for this group. For individuals with an elevated GMI, the PPG was consistently lower at lunch (30 ± 20%), dinner (36 ± 23%), and snacks (34 ± 23%) in comparison to FPG. There was no significant difference between PPG and FPG during breakfast.

People who used mealtime insulin did not show a different pattern of the daily and mealtime relative contribution of the FPG and PPG compared to people who did not use mealtime insulin.

### 3.3. Carbohydrate Consumption

Carbohydrate consumption was highest during dinner, averaging 54.6 [43.0–69.0] grams. In contrast, the lowest intake was recorded at breakfast, with 34.4 [24.6–45.0] grams (*p* < 0.001, see Table 2 for details).

No significant correlations were identified between overall daily PPG and daily carbohydrate intake. Similarly, post-lunch, post-dinner, and post-snack carbohydrate intakes did not exhibit a correlational relationship with PPG. However, a moderate positive correlation (r = 0.41, *p* < 0.01 post-Holm–Bonferroni correction) was observed after breakfast, indicating that higher carbohydrate consumption during this meal is associated with an increased PPG.

## 4. Discussion

A main finding of the study is that the relative contribution of postprandial hyperglycemia becomes more relevant in people with better glycemic control (HbA_1c_ or GMI < 7.0%, 53 mmol/mol). However, even in those with poorly controlled blood glucose levels, the contribution of PPG, although relatively lower, is still high with 49.2% for GMI and 60.5% for HbA_1c_. It is therefore evident that PPG is an important parameter across all levels of glycemic control.

Conversely, the relative contribution of FPG decreases with improved glycemic control, showing the importance of reducing fasting glucose levels to enhance overall glucose management in patients with higher HbA_1c_ or GMI levels. This observation aligns with existing literature and offers more depth compared to earlier studies that relied on SMBG data rather than CGM data [9,10].

We identified significant correlations between PPG and HbA_1c_, GMI, and TAR. Additionally, there were also correlations between FPG and HbA_1c_, GMI, and TAR, highlighting that both PPG and FPG are parameters reflecting glucose regulation. These findings are consistent with prior literature that also reported correlations between HbA_1c_ and PPG and FPG [11,20,21]. Notably, the correlation between FPG and HbA_1c_ was lower than the correlation between FPG and GMI, which has not been previously explored. This is not unexpected considering that GMI is a CGMbased parameter derived from the same data source as PPG and FPG. 

Obviously, controlling fasting glucose levels is essential across all circumstances, especially during periods of overall poor glycemic control. Our study indicates the potential utility of daily PPG as an informative parameter for assessing the daily nutritional impact on glycemia. This parameter could prove valuable as a treatment target, as it can be easily and effectively utilized in clinical practice to gain a comprehensive understanding of how medication usage and carbohydrate intake affect diabetes management, potentially without requiring detailed knowledge of a patient’s specific dietary habits.

Additionally, mealtime PPG can be a valuable parameter in daily clinical practice, providing insights into which specific meals contribute most to an individual patient’s hyperglycemia. The relative PPG contribution is higher after breakfast compared to other meals, potentially reinforced by the dawn phenomenon [22]. This suggests that, for most patients, it is particularly important to focus on reducing PPG caused by breakfast. Interestingly, using the outcome measure of PPG could address the absence of dietary information for those who find carbohydrate counting too laborious or focus solely on one meal instead of counting carbohydrates for all meals. This emphasizes the practical relevance of considering mealtime PPG as a useful tool for tailoring diabetes management strategies [12].

Furthermore, a correlation was found between PPG and the amount of carbohydrates in a meal. Nevertheless, the highest increase in blood glucose was observed after breakfast, despite patients consuming fewer carbohydrates during this meal compared to lunch and dinner. This might be caused by the type of carbohydrates, causing a steeper and higher rise whereas the total hyperglycemia was not different compared to other meals.

Some methodological considerations had to be made. The relative contribution of PPG and FPG is partly determined by the method used to calculate both PPG and FPG. In particular, the FPG has a significant impact on the calculation. A two-hour PPG window was used in this study, and the absence of a fixed diet regimen resulted in overlapping mealtimes that could not be included in the analysis. Additionally, a clinically relevant glucose rise of 0.5 mmol/L after consumption was assumed to determine the minimum amount of carbohydrates causing a glucose rise, as this had not been previously established.

Several potential limitations of this study need to be addressed. Dietary intake was self-reported using food diaries, which may introduce subjective bias and inaccuracies in estimating intake quantities. Although compliance with diary recording was high, omissions and inaccuracies are possible. Furthermore, the study utilized data exclusively from the FreeStyle Libre 1 sensor, and findings may not generalize to other devices. The study cohort comprised patients with type 2 diabetes treated in a Dutch hospital, raising questions about the generalizability of results to diverse populations with different demographics, disease profiles, and device accuracy levels.

Additionally, all participants in this study were under diabetes medication and aimed to maintain stable glucose levels, irrespective of their lifestyle habits, influencing their PPG and FPG values. However, this study is the first to explore the relationship between fasting and postprandial glucose levels and hyperglycemia using real-world data, reflecting the daily lives of patients. They were not influenced by their glucose values, as the glucose sensor was blinded. Moreover, this study uniquely integrated continuous glucose monitoring (CGM) data with dietary information, strengthening the overall results.

## 5. Conclusions

In conclusion, this real-world study underscores the practical importance of considering both FPG and PPG, particularly after breakfast, in diabetes management in patients with type 2 diabetes. The integration of CGM data with dietary information offers a comprehensive perspective, allowing for tailored treatment strategies. These insights can enhance the care and treatment of individuals with type 2 diabetes, ultimately contributing to better glycemic control and improved patient outcomes.

## Figures and Tables

**Figure 1 nutrients-16-03557-f001:**
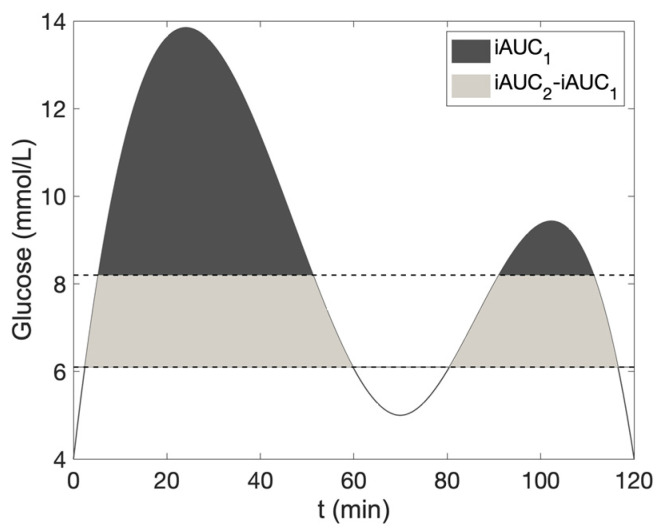
Postprandial plasma glucose (PPG), represented by the incremental area under the curve above the fasting plasma glucose (FPG) level (iAUC1). The FPG contribution is represented by the area where the values are greater than 6.1 mmol/L and less than the FPG level (iAUC2-iAUC1). In this example, both FPG and PPG are computed within the timeframe of 0 to 120 min following a meal, as depicted on the *x*-axis. This methodology is also applied to compute daily FPG and PPG contributions within the timeframe of 06:00 to 22:00.

**Figure 2 nutrients-16-03557-f002:**
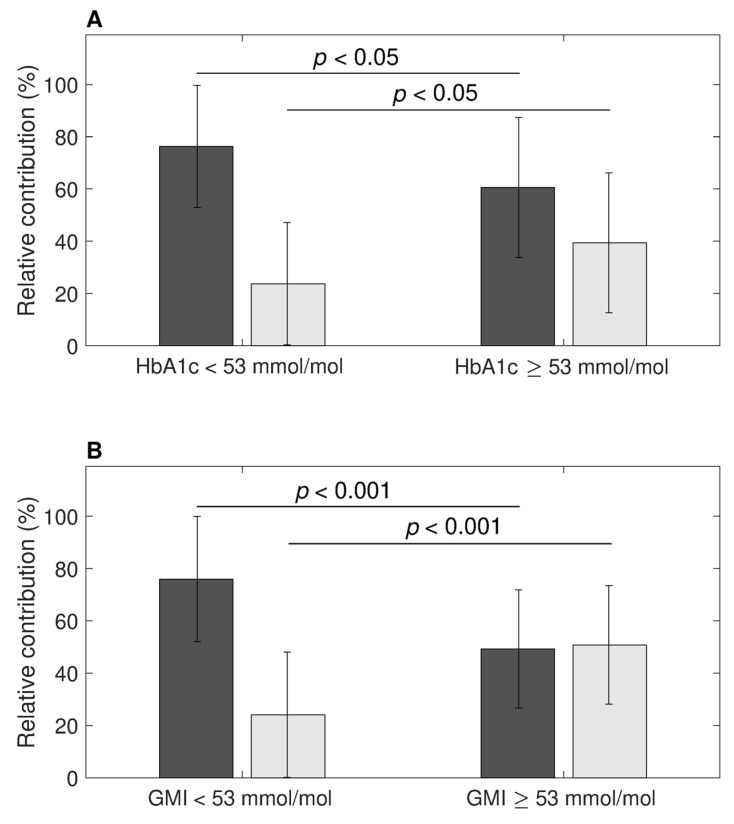
The relative contributions of postprandial (dark grey) and fasting (light grey) hyperglycemia as percentages to the overall daily hyperglycemia across two ranges of HbA_1c_ (**A**) and GMI (**B**).

**Figure 3 nutrients-16-03557-f003:**
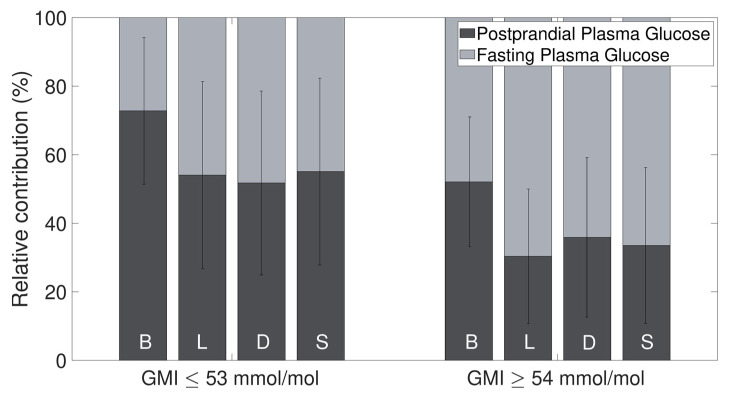
Relative contributions of postprandial plasma glucose (PPG) and fasting plasma glucose (FPG) after breakfast (B), lunch (L), dinner (D), and snacks (S), respectively from left to right, to GMI values on target (<53 mmol/mol, 7.0%) and above target (≥53 mmol/mol, 7.0%). The standard deviations are represented by the error bars.

**Table 1 nutrients-16-03557-t001:** Patient characteristics.

Characteristic	*n*	Value
Age, years	59	65.7 ± 10.0
Gender, men *n* (%)	59	39 (66)
Diabetes duration, years	59	14.2 ± 10.7
HbA_1c_, %	59	7.4 ± 3.0
HbA_1c_, mmol/mol	59	56.9 ± 9.1
BMI, kg/m^2^	59	31.0 ± 5.2
Insulin use, *n* (%) yes	59	35 (59)
Multiple daily insulin injection, *n* (%) yes	59	25 (42)

Abbreviation: HbA_1c_, glycated hemoglobin.

**Table 2 nutrients-16-03557-t002:** Measures describing postprandial plasma glucose 2 h after different mealtimes.

**Measure**	Breakfast (B)	Lunch (L)	Dinner (D)	Snack (S)	*p*	Post Hoc Test
iAUC_1_, mmol/L·min	234.0 [149.4–314.1]	115.6 [76.3–232.4]	155.3 [73.5–272.2]	132.8 [78.9–227.3]	<0.001	B> L, S
Δglucose, mmol/L	3.6 ± 1.4	2.5 ± 1.6	2.4 ± 1.2	2.0 ± 1.1	<0.001	B> L, D, S
Slope, mmol/L/min	0.05 ± 0.02	0.03 ± 0.02	0.03 ± 0.02	0.02 ± 0.02	<0.001	B> L, D, S
Peak time, min	81.6 ± 21.4	83.2 ± 17.9	77.0 ± 17.4	76.2 ± 13.3	0.08	
Peak, mmol/L	11.0 ± 2.2	10.1 ± 2.0	10.3 ± 2.1	9.9 ± 1.9	0.09	
Carbohydrate intake, g	34.4 [24.6–45.0]	41.1 [35.3–52.0]	54.6 [43.0–69.0]	33.3 [26.5–43.9]	<0.001	D> B, L, S & L> B, S

Abbreviation: iAUC, incremental area under the curve.

## Data Availability

Data are available upon reasonable request after the inclusion of the patients has been completed.

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
