# Peer review of "The Contribution of Postprandial Glucose Levels to Hyperglycemia in Type 2 Diabetes Calculated from Continuous Glucose Monitoring Data: Real World Evidence from the DIALECT-2 Cohort"

_nutrients, 2024, doi:10.3390/nu16203557_

Round 1
Reviewer 1 Report
Comments and Suggestions for Authors
Dear Authors:
Regarding the manuscript with title “The Contribution of Postprandial Glucose Levels to Hyperglycemia in Type 2 Diabetes Calculated From Continuous Glucose Monitoring Data: Real World Evidence From the DIALECT-2 Cohort”, I have several minor comments to address.
Comment 1:
Why authors direct this manuscript only to hyperglycemia and not also to hypoglycemia, as continuous glucose monitoring provides information regarding TAR and TBR?
Comment 2:
Why authors refer on title only to PPG and not also to FPG. I refer this question according to objective and conclusion of the study.
Comment 3:
On Abstract, authors must insert a paragraph regarding “Background”.
Comment 4:
On Abstract, I suggest authors to add the main objective of this study, that according to what is stated on lines 64-65 is to “determine the correlations and relative contribution of FPG and PPG to overall hyperglycemia in patients with type 2 diabetes”
Comment 5:
On Abstract, on the subchapter of Conclucions, authors must give response to the main objective of the study.
Comment 6:
Line 66-68: “Additionally, we aim to assess these contributions after different meals to gain a comprehensive understanding by analyzing the differences in relative contribution of FPG and PPG after specific meals.”. In this sentence, it makes no sense to refer to the relative contribution of FPG after specific meals, as FPG refers is measured after a period of fasting. Thus, I suggest authors to rephrase the previous sentence.
Comment 7:
Lines 66-69: “Additionally, we aim to assess these contributions after different meals to gain a comprehensive understanding by analyzing the differences in relative contribution of FPG and PPG after specific meals. Through this study, we seek to explore the relationship between glucose management and mealtime patterns in patients with type 2 diabetes.” I kindly ask authors what is the difference between the contente of these two sentences, as it appears to me that they contain similar content. Thus, in my opinion, authors can withdrawn the second sentence.
Comment 8:
Line 71: Regarding patient inclusion, as authors refer to exclusion criteria, it is also important to refer inclusion criteria.
Comment 9:
I suggest authors to number the subchapter “Data collection”. Thus, the other subchapters of Methods must be renumbered”. “2.3. “Data coillection”; “2.4. “Data analysis”; “2.5. “Statistical analysis”
Comment 10:
I suggest authors to add a Table with participants’ characteristics.
Comment 11:
I kindly ask authors to explain lines 274-276.
Comment 12:
Lines 279-281 must be transferred to the final of the chapter of Discussion when authors must refer the strengths of the present study.
Comment 13:
Line 284: I suggest authors to change “even in those with not well-controlled glycemia” by “even in those with poorly controlled blood glucose levels”.
Comment 14:
Line 349: I suggest authors to change “of individuals with diabetes” by “of individuals with type 2 diabetes”
Author Response
Comment 1: Why authors direct this manuscript only to hyperglycemia and not also to hypoglycemia, as continuous glucose monitoring provides information regarding TAR and TBR?
Response 1: Thank you for your comment. The focus of this manuscript is primarily on hyperglycemia because we aim to investigate how dietary factors contribute to hyperglycemic episodes. However, we acknowledge that the effects of continuous glucose monitoring on hypoglycemia are indeed interesting and warrant exploration in future studies. We appreciate your suggestion and will consider including this aspect in subsequent research.
Comment 2: Why authors refer on title only to PPG and not also to FPG. I refer this question according to objective and conclusion of the study.
Response 2: Thank you for your question. The title focuses solely on PPG to keep it concise, as the main emphasis of the study is on postprandial glucose. We made this choice to maintain clarity, given that the title was already quite long. However, we acknowledge the importance of FPG in the study and discuss its role in the objectives and conclusions.
Comment 3: On Abstract, authors must insert a paragraph regarding “Background”.
Response 3: Thank you for your feedback. I have now added a paragraph regarding the “Background” in the abstract to provide better context for the study.
Comment 4: On Abstract, I suggest authors to add the main objective of this study, that according to what is stated on lines 64-65 is to “determine the correlations and relative contribution of FPG and PPG to overall hyperglycemia in patients with type 2 diabetes”
Response 4: Thank you for your suggestion. I have updated the abstract to include the main objective of the study, as stated in lines 64-65.
Comment 5: On Abstract, on the subchapter of Conclucions, authors must give response to the main objective of the study.
Response 5: Thank you for your comment. I have now revised the conclusion in the abstract to directly address the main objective of the study, reflecting the findings on the contributions of FPG and PPG to hyperglycemia.
Comment 6: Line 66-68: “Additionally, we aim to assess these contributions after different meals to gain a comprehensive understanding by analyzing the differences in relative contribution of FPG and PPG after specific meals.”. In this sentence, it makes no sense to refer to the relative contribution of FPG after specific meals, as FPG refers is measured after a period of fasting. Thus, I suggest authors to rephrase the previous sentence.
Response 6: Thank you for your suggestion. I have rephrased the sentence to: "Additionally, we aim to assess these contributions after different meals to gain a comprehensive understanding by analyzing the differences in the relative contribution of FPG and PPG." This adjustment removes the unclarity and duplication in the original sentence.
Comment 7: Lines 66-69: “Additionally, we aim to assess these contributions after different meals to gain a comprehensive understanding by analyzing the differences in relative contribution of FPG and PPG after specific meals. Through this study, we seek to explore the relationship between glucose management and mealtime patterns in patients with type 2 diabetes.” I kindly ask authors what is the difference between the contente of these two sentences, as it appears to me that they contain similar content. Thus, in my opinion, authors can withdrawn the second sentence.
Response 7: Thank you for your helpful comment. We have merged the two sentences to avoid repetition.
Comment 8: Line 71: Regarding patient inclusion, as authors refer to exclusion criteria, it is also important to refer inclusion criteria.
Response 8: We have now clarified the inclusion criteria in the manuscript, as suggested.
Comment 9: I suggest authors to number the subchapter “Data collection”. Thus, the other subchapters of Methods must be renumbered”. “2.3. “Data coillection”; “2.4. “Data analysis”; “2.5. “Statistical analysis”
Response 9: We agree that this should indeed be a separate subchapter. We have now made the necessary adjustments and renumbered the subchapters accordingly in response to your feedback.
Comment 10: I suggest authors to add a Table with participants’ characteristics.
Response 10: We have added a table with participants’ characteristics as requested.
Comment 11: I kindly ask authors to explain lines 274-276.
Response 11: Thank you for your observation. This was standard text from the Nutrients template that I accidentally did not remove. It has now been removed.
Comment 12: Lines 279-281 must be transferred to the final of the chapter of Discussion when authors must refer the strengths of the present study.
Response 12: We have transferred the content from lines 279-281 to the end of the Discussion chapter.
Comment 13: Line 284: I suggest authors to change “even in those with not well-controlled glycemia” by “even in those with poorly controlled blood glucose levels”.
Response 13: I have revised the text and changed "even in those with not well-controlled glycemia" to "even in those with poorly controlled blood glucose levels" as recommended.
Comment 14: Line 349: I suggest authors to change “of individuals with diabetes” by “of individuals with type 2 diabetes”
Response 14: Thank you for your suggestion. I have updated the text, changing "of individuals with diabetes" to "of individuals with type 2 diabetes" as recommended.
Reviewer 2 Report
Comments and Suggestions for Authors
The manuscript “The Contribution of Postprandial Glucose Levels to Hypergly-2 cemia in Type 2 Diabetes Calculated From Continuous Glucose 3 Monitoring Data: Real World Evidence From the DIALECT-2 4 Cohort” is very interesting, is well written, in a clearer and concise manner, and is scientifically sound.
- I suggest that the authors include a brief background in the abstract of the work, since they only handle the Objective, Materials and Methods, Results and Conclusions.
- Line 275-275: please remove “This section may be divided by subheadings. It should provide a concise and precise description of the experimental results, their interpretation, as well as the experimental conclusions that can be drawn.”
Author Response
Comments 1: I suggest that the authors include a brief background in the abstract of the work, since they only handle the Objective, Materials and Methods, Results and Conclusions.
Response 1: Thank you for the suggestion. I have now incorporated a brief background into the abstract to provide better context for the study, addressing the gap you mentioned.
Comments 2: Line 275-275: please remove “This section may be divided by subheadings. It should provide a concise and precise description of the experimental results, their interpretation, as well as the experimental conclusions that can be drawn.
Response 2: Thank you for pointing this out. I have removed this sentence.